

# Vegetation indices' spatial prediction based novel algorithm for determining tsunami risk areas and risk values

Kristoko Dwi Hartomo[1], Yessica Nataliani[1] and Zainal Arifin Hasibuan[2]

[1] Department of Information System, Faculty of Information Technology, Satya Wacana Christian University, Salatiga, Indonesia
[2] Faculty of Computer Science, University of Dian Nuswantoro, Semarang, Indonesia

## ABSTRACT

This paper aims to propose a new algorithm to detect tsunami risk areas based on spatial modeling of vegetation indices and a prediction model to calculate the tsunami risk value. It employs atmospheric correction using DOS1 algorithm combined with $k$-NN algorithm to classify and predict tsunami-affected areas from vegetation indices data that have spatial and temporal resolutions. Meanwhile, the model uses the vegetation indices (*i.e.*, NDWI, NDVI, SAVI), slope, and distance. The result of the experiment compared to other classification algorithms demonstrates good results for the proposed model. It has the smallest MSEs of 0.0002 for MNDWI, 0.0002 for SAVI, 0.0006 for NDVI, 0.0003 for NDWI, and 0.0003 for NDBI. The experiment also shows that the accuracy rate for the prediction model is about 93.62%.

# INTRODUCTION

Tsunami is one of the disaster threats for many coastal areas in Indonesia. This disaster is generally triggered by an earthquake at sea causing a vertical shift in the seabed (*Amri et al., 2018*). The territories of Indonesia are surrounded by the meeting of the world's main tectonic plates which causes the tsunami. These plates are: (1) the Indian-Australian Ocean plate in the south, moving relatively to the north and presses the Eurasian plate (where most of Indonesia territories are located), and (2) the Pacific plate in the east, which moves relatively westward against the Eurasian plate. This phenomenon causes many sources of earthquakes as well as the growth of active volcanoes in the territories of Indonesia, thus placing Indonesia as one of the areas in the world's most active tectonic zone (*Verstappen, 2010*).

The National Development Planning Agency of Indonesia (Bappenas) stated that the total loss and damage from the tsunami and earthquake in Yogyakarta and Aceh Province was Rp 70.5 trillion. The impacts were 80% of the infrastructure sector damage (including housing) and 11% of the productive sector damage (Regulation of the Ministry of Health Republic of Indonesia No. 36 of 2014 about Assessment of Post-Disaster Damage, Loss and Health Resource Needs, 2014). The tsunami itself caused the death toll

Corresponding author
Kristoko Dwi Hartomo,
kristoko@uksw.edu

in the affected area reaching 108,100 people and 127,700 people missing (*Amri et al., 2018*).

The province of Yogyakarta that has the potential for a tsunami is located in the south coast area. The disaster risk index in Yogyakarta for the tsunami disaster currently reaches 1.74. This value is seen from the vulnerability of the Yogyakarta region from the tsunami disaster by considering geological aspects, tsunami threat maps and demographics in villages that may be exposed to the impact of the tsunami. In general, the Yogyakarta region has a fairly high tsunami disaster index, especially areas directly adjacent to the coastline (*Regional Disaster Management Agency, 2019*). Kulon Progo Regency is the most susceptible to tsunami disaster among the four districts in Yogyakarta. There is a possibility of 60,607 people that will be affected if a tsunami occurs in this district (*National Disaster Management Agency, 2012*). Areas with a high tsunami risk in Kulon Progo Regency are Temon Sub-District, Wates Sub-District, Panjatan Sub-District, Lendah Sub-District, and Galur Sub-District (Regulation of the Special Region of Yogyakarta No. 5 of 2019 about The Spatial Plan of the Special Region of Yogyakarta for 2019–2039, 2019). Kulon Progo Regency comprises of 47 villages (see Fig. 1 for further information on the name of the villages) (*Mustaqim, 2019*). Figure 1 shows the visualization of the map of Kulon Progo Regency. Figure 1 was obtained from the mapping of Shuttle Radar Topography Mission (SRTM) image from https://earthexplorer.usgs.gov/.

Damage and impacts caused by disasters, especially tsunamis, can be analyzed using the field survey method. The field survey method used during a disaster is difficult and carries a high risk. Disasters can occur over a large area, so surveying the entire affected area takes a long time (*Singh, Nigam & Pal, 2014*). Obtaining fast and accurate information in the event of a disaster can save many human lives and reduce losses. The solution for problems in field survey is the use of satellite technology. With the availability of easily obtained satellite images, remote sensing techniques can be widely used to assess disaster risk areas (*Brunner, Lemoine & Bruzzone, 2010*). The use of satellite imagery can overcome the limitations of collecting disaster area image data using traditional methods. Satellite images not only speed up the process of analyzing disaster areas but also provide accurate and timely estimates of the risk of disaster affected areas (*Koshimura et al., 2020*).

Landsat 8 satellite captures land cover images in disaster-affected areas that are influenced by soil and vegetation characteristics, in which this process becomes the basis for remote sensing techniques in analyzing land cover (*Rendana et al., 2016*). In the last five years, the vegetation indices analysis has been applied in the evaluation of land cover. One of them is the analysis of disaster-affected land as a variation of spatial and temporal analysis techniques (*Holzman, Rivas & Bayala, 2014*; *Mallick, Bhattacharya & Patel, 2009*). Digital image processing from satellite data enables image analysis through various algorithms and mathematical indices. Meanwhile, feature analysis is based on the reflectance characteristics and the index has been designed to detect prominent features in the image area (*Xie et al., 2010*). There are several indices that can detect areas containing vegetation in the image obtained from remote sensing, such as Normalized Difference Vegetation Index (NDVI), Normalized Difference Built-Up Index (NDBI), Soil

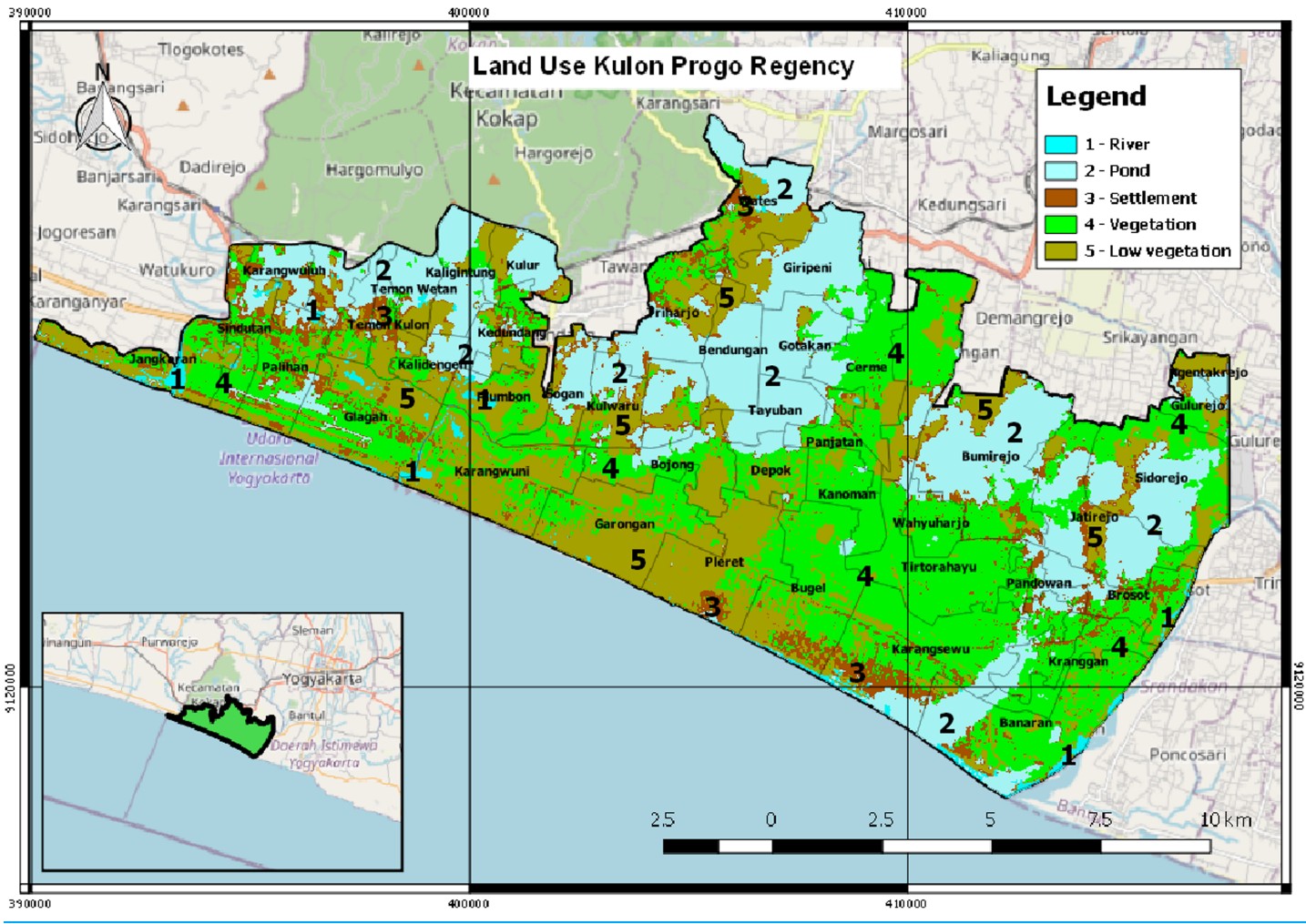

**Figure 1 The visualization of Kulon Progo regency map.**

Adjusted Vegetation Index (SAVI), Normalized Difference Water Index (NDWI), and Modified Normalized Difference Water Index (MNDWI). NDVI is a common and widely used index (*Karnieli et al., 2010*). It is also applied considerably in global environment, climate change, and disasters research (*Gao, 1996*).

The response of vegetation to the environment is considered very sensitive. It affects the ecological and climate balances. In addition it becomes an effective barrier against natural disasters. Classification of natural disaster images, especially images before and after the tsunami, can use the average value of various spectrum indices such as NDVI, NDBI, SAVI, MNDWI, and NDWI as training data (*Singh, Nigam & Pal, 2014*). The pre- and post-tsunami images are compared to obtain the inundated area. NDVI, NDBI, SAVI, MNDWI, and NDWI are calculated as a ratio difference between red and near infrared bands respectively in measured canopy reflectance (*Hu et al., 2008*). The process of analyzing the vegetation index data is carried out using representation, categorization, and classification approaches to produce valid information. The most

suitable method to represent, categorize, and classify the vegetation index is a machine learning (*Maxwell, Warner & Fang, 2018*).

The most recent development of remote sensing methods for impact analysis and detection of image transformations caused by disasters is the use of machine learning. The machine learning-based transformation detection method consists of two categories, they are supervised and unsupervised detection methods. The supervised method requires training to identify transformations, such as methods of instance support vector machine (SVM) (*Bovolo & Bruzzone, 2007*; *Volpi et al., 2012*), post classification comparison (*Yang & Wen, 2011*), artificial neural network (ANN), and function neural network (FNN) (*Mehrotra et al., 2015*). Meanwhile, the unsupervised method does not require training as this method analyzes the image to identify any transformations. The unsupervised method is usually used to differentiate images (*Wang et al., 2020*) and make a principal component analysis (PCA) (*Corner, Narayanan & Reichenbach, 2000*).

In this study, a new algorithm is proposed to predict tsunami risk areas based on spatial modelling prediction of vegetation index. The new algorithm is developed with the addition of atmospheric correction before the data is processed using the $k$-Nearest Neighbor ($k$-NN) algorithm. In this algorithm, machine learning is used to classify and predict tsunami-affected areas from vegetation indices data that has spatial and temporal resolutions. To evaluate the performance of this new algorithm, it is compared to Cart, SVM, and ANN algorithms. The experimental results show that this new framework is superior, as indicated by the Mean Squared Error (MSE), Root Mean Squared Error (RMSE), Mean Absolute Error (MAE), and Cohen's Kappa scores that outperform other algorithms. Furthermore, since there is only a category of risk level and no value that indicates the level of tsunami risk, then a new formula using vegetation indices and other parameters is proposed to determine the tsunami risk value.

This paper is presented as follows. "Introduction" describes the background on tsunami problems. "Related Works" contains some related works and the reviews of machine learning and vegetation indices. "Research Methods" presents the research method and flowchart, as well as the proposed algorithm. A new algorithm for detecting tsunami risk areas is proposed, by adding atmospheric correction, before applying prediction and classification algorithms, since some data may contain some noises. This framework uses $k$-NN algorithm to classify the area, with high, medium, and low risk. "Results and Discussions" explains the experiments and comparisons of the proposed framework using $k$-NN algorithm with other classifying methods using real data. A model to predict the tsunami risk value is also discussed in this section. Finally, the conclusions are stated in "Conclusions".

## RELATED WORKS

In this section, some research about remote sensing based on prediction and classification are discussed. *Prasetyo et al. (2020)* implemented spectral vegetation index to the data obtained from the Landsat 8 OLI satellite to provide disaster risk index information. $k$-NN with spatial autocorrelation was used to classify the drought risk areas. The spectral vegetation indices used in this study were NDVI, SAVI, Vegetation Condition Index

(VCI), Temperature Condition Index (TCI), and Vegetation Health Index (VHI) (*Xue & Su, 2017*). While, the Kappa accuracy test showed that the SVM and *k*-NN methods had an accuracy of 88.30.

Other research were conducted a remote sensing study using machine learning methods for classifying the image data (*Ma et al., 2019*; *Maxwell, Warner & Fang, 2018*). One of them wanted to classify the distribution, species, and extent of mangroves using the Akaike Information Criterion (AIC) on Nusa Lembongan Island (*Ilham & Marzuki, 2017*). This research was important since mangrove forest had important ecological, economic, and social roles. One example was the study on mangrove forest as a green belt for shoreline protection from storms and tsunami waves using Worldview-2 satellite imagery with a data resolution of 0.46 m. This method automatically identified land classes, sea/water classes, and mangrove classes. The results showed that the classification accuracy was 68.32%.

Another related study proposed a new method for classifying daily NDVI time series data based on a combination of multi-classifiers (*Zhao et al., 2017*). In this study, the HJ–CCD satellite was used as data for compiling an NDVI time series model with S–G filtering and spatial interpolation. This study also proposed a dimension reduction method using the statistical features of the daily NDVI time series. The results showed that the accuracy of image classification of disaster-prone areas was 77.45%.

The next study used the unsupervised classification method to classify coastal forest damage due to tsunamis (*Inoue & Yonezawa, 2015*). The extent and distribution of tsunami damage was predicted using NDVI. To assess the classification accuracy, they used the error matrix. Aerial photos were used for reference. From 200 random points, the accuracy was 79.50% with a Kappa Stats of 0.6387.

Similar research proposed NDVI predictions recorded by satellite at Ventspils City in Courland, Latvia and obtained using the Markov chain method (*Stepchenko & Chizhov, 2016*). In general, Markov chain prediction is a probability forecasting method because the prediction results show the probability of an NDVI value in the future (*Liu, 2010*). This study demonstrated how Markov chains could predict future values with less memory and random walk capability. Each state was reached directly by other states with a transition matrix to provide a high prediction accuracy of 63.93%.

Subsequent research proposed a method for mapping the value of environmental damage after a disaster (*Havivi et al., 2018*). The data used are TerraSAR-X (TSX) images with high resolution obtained from before and after the incident and also Landsat 5 images before the incident. The affected areas were analyzed with Synthetic Aperture Radar (SAR) using one SAR interferometric coherence map (InSAR). The accuracy of mapping the environmental damage caused by the tsunami could be improved using the vegetation index (NDVI) (*Ghebrezgabher et al., 2020*; *Koshimura et al., 2020*). The affected areas were mapped with the overall accuracy of 89% and Kappa coefficient of 82%.

In summary, it has been shown from this review that spectral vegetation indices indicated the quantitative values for measuring the vegetation canopy in receiving and reflecting the light spectrum. They were interpreted as plant's spectral characteristics,

including the infrared spectrum as visible light (IR) and the near infrared spectrum as invisible light (NIR) (*Prasetyo et al., 2020*).

## RESEARCH METHODS

In this section, a research method for detecting the tsunami risk areas is introduced. Atmospheric correction is used in this algorithm since data may contain some noises which will interfere with the prediction and classification performance. The flowchart for this research method is depicted in Fig. 2 and can be explained as follows.

Figure 2 can be explained as below.

1. Data extraction

   Landsat 8 image data that has been downloaded from https://earthexplorer.usgs.gov/ in the form of digital values can be used for land use mapping. The extracted data consists of 11 bands, namely Band 1, Band 2, …, Band 11, each with a different name and wavelength. Each band has a wavelength between 0.43 and 12.51. Table 1 shows the Landsat 8 operational land image (OLI) and thermal infrared sensor (TIRS).

2. Atmospheric correction

   The data obtained from data extraction still has a low level of radiometric accuracy because it has errors in the recording process from the sensor in the image. Therefore, if the data will be used for data processing such as biomass, vegetation indices, land-cover classification, and so on, they require an atmospheric correction process. This process is used to improve the accuracy of image classification so that the data obtained can be compared and arranged in a number of solutions and evaluation of tsunami-prone areas. One of the processes is the removal of atmospheric haze and cloud cover. The results of the atmospheric correction process are very important for optimizing foggy satellite imagery regarding the object's surface to detect changes in land cover and land use.

   The method that can be used in the atmospheric correction process is the Dark Object Subtraction (DOS), especially DOS1. This method is chosen because the field data parameters for image correction are not known and also the atmospheric effect model is not known which shows the condition of an image when the image is recorded (*Zhang, He & Wang, 2010*).

$$\rho = \frac{\pi \left( L_{sat} - L_p \right) d^2}{E_0 \cos(\theta_z)} \tag{1}$$

where $L_{sat}$ is the at-satellite radiance, $L_p$ is the path radiance, $\rho$ is the land surface reflectance, $d$ is the Earth–Sun distance in astronomical units, $E_0$ is the exoatmospheric solar spectral irradiance, and $\theta_z$ is the solar zenith angle.

3. Data pre-processing

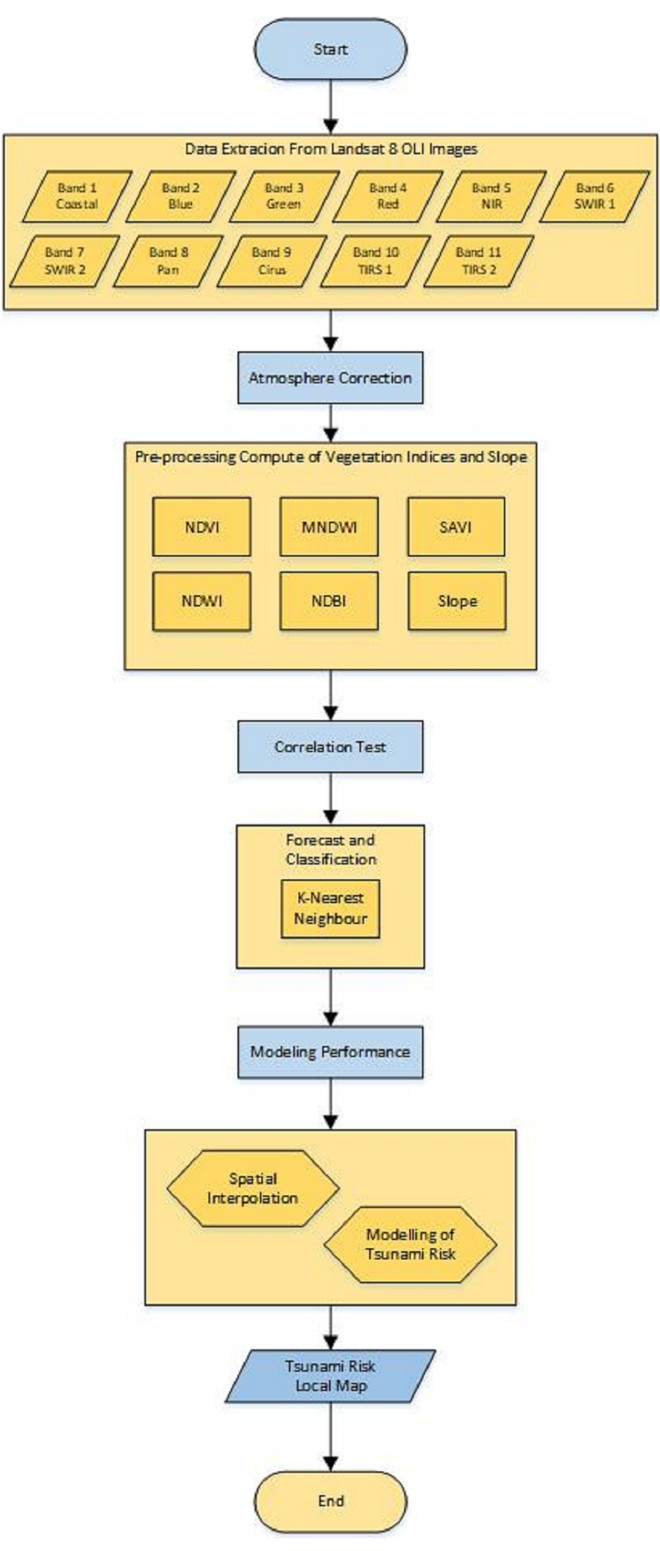

**Figure 2  Flowchart of the research method.**

**Table 1 Landsat 8 operational land image (OLI) and thermal infrared sensor (TIRS) (*U.S. Geological Survey, 2021*).**

| Band name | Band | Bandwidth | Resolution |
|---|---|---|---|
| Band 1 | Coastal aerosol | 0.43–0.45 | 30 |
| Band 2 | Blue | 0.45–0.51 | 30 |
| Band 3 | Green | 0.53–0.59 | 30 |
| Band 4 | Red | 0.64–0.67 | 30 |
| Band 5 | NIR (Near Infrared) | 0.85–0.88 | 30 |
| Band 6 | SWIR (Short-wave Infrared) 1 | 1.57–1.65 | 30 |
| Band 7 | SWIR (Short-wave Infrared) 2 | 2.11–2.29 | 30 |
| Band 8 | Panchromatic | 0.50–0.68 | 15 |
| Band 9 | Cirrus | 1.36–1.68 | 30 |
| Band 10 | TIRS 1 | 10.60–10.19 | 100 |
| Band 11 | TIRS 2 | 11.50–12.51 | 100 |

To begin with, the geometric and radiometric corrections need to be known. Geometric correction is used to correct the position of the coordinates of each pixel in the image exactly as displayed on the surface of the earth. It involves satellite movements, earth rotation, and terrain effects. Meanwhile, radiometric correction has a different goal. The stage of data pre-processing includes collecting Sentinel 2 satellite image data obtained from www.earthexplorer.usgs.gov. They are corrected not only geometrically, but also radiometrically and atmospherically.

After the image is corrected, the clean band is calculated, according to the formula for each index. Sentinel 2 image data extraction uses the NDVI, NDBI, NDWI, MSAVI, and MNDWI formulas. The extraction results are numerical values, which can be used for the classification and prediction using $k$-NN. NDVI, MNDWI, MSAVI, NDWI, and NDBI formulas are explained as follows.

- Normalized Difference Vegetation Index (NDVI)

NDVI is a vegetation index that is often used to compare the level of vegetation greenness (chlorophyll level) in plants (*Min, Bahar & Udin, 2016*).

$$NDVI = \frac{NIR - Red}{NIR + Red} = \frac{Band\ 5 - Band\ 4}{Band\ 5\ + Band\ 4} \tag{2}$$

- Normalized Difference Water Index (NDWI)

NDWI is an index that shows the wetness level of an area. The formula for the NDWI index follows Eq. (3) (*Xu, 2006*).

$$NDWI = \frac{Green - NIR}{Green + NIR} = \frac{Band\ 3 - Band\ 5}{Band\ 3 + Band\ 5} \tag{3}$$

- Modified Normalized Difference Water Index (MNDWI)

MNDWI is a modification of the NDWI index. The formula for MNDWI index follows Eq. (4) (*Acharya, Subedi & Lee, 2018*).

$$MNDWI = \frac{Green - SWIR\ 1}{Green + SWIR\ 1} = \frac{Band\ 3 - Band\ 6}{Band\ 3\ + Band\ 6} \tag{4}$$

- Soil Adjusted Vegetation Index (SAVI)

SAVI is used to correct NDVI for minimizing the influence of soil brightness in low vegetative cover areas, using a soil-brightness correction factor ($L$). SAVI is calculated as a ratio between the Red and NIR values with a soil brightness correction factor, where $L = 0.5$, to accommodate most land cover types. The formula for SAVI index follows Eq. (5) (*Huete, 1988*).

$$SAVI = \left(\frac{NIR - Red}{NIR + Red + L}\right) * (1 + L) = \left(\frac{Band\ 5 - Band\ 4}{Band\ 5\ + Band\ 4 + 0.5}\right) * 1.5 \tag{5}$$

- Normalized Difference Built-up Index (NDBI)

NDBI is an effective transformation/index used in mapping building lands in an area automatically using Landsat 8 OLI images. The formula for NDBI index follows Eq. (6) (*Zha, Gao & Ni, 2003*).

$$NDBI = \frac{SWIR\ 1 - NIR}{SWIR\ 1\ + NIR} = \frac{Band\ 6 - Band\ 5}{Band\ 6\ + Band\ 5} \tag{6}$$

4. Correlation test

Correlation test calculates the correlation between variables and analyzes the level of closeness of the relationship between the independent variable ($X$) and the dependent variable ($Y$). Pearson correlation is one of the correlation tests used to measure the strength of the linear relationship between two variables. Two variables are said to be correlated if a change in one variable is accompanied by a change in another variable, both of which change in the same direction or vice versa. The formula for calculating correlation can be seen in Eq. (7).

$$r_{xy} = \frac{n \sum XY - (\sum X)(\sum Y)}{\sqrt{\left\{n \sum X^2 - (\sum X)^2\right\}\left\{n \sum Y^2 - (\sum Y)^2\right\}}} \tag{7}$$

where $r_{xy}$ is the correlation value, $X$ is the variable $X$, and $Y$ is the variable $Y$.

5. Forecast and classification

Supervised classification is used to classify image data and form a thematic map of Land Used Land Cover of the study area. Contour analysis of the study area is the goal behind the classification of the Shuttle Radar Topographic Mission (SRTM) Digital Elevation

Model (DEM) image data. $k$-NN is the classification method for a set of data based on pre-existing learning data. The classification process is done by finding the closest point from the old point of "a" to a new point of "a" (nearest neighbor). The closest point search technique is performed by using the Euclidean distance formula, as shown in Eq. (8). To use the $k$-NN algorithm, it is necessary to determine the value of $k$, where $k$ is the number of nearest neighbors used to classify the new data.

$$d(x, y) = \sqrt{\sum_{i=1}^{n} (x_i - y_i)^2} \tag{8}$$

where $n$ is the number of data.

6. Spatial Interpolation

Interpolation is the method used to predict a value at locations where data are not available. Thus, interpolation is used to predict values of the surrounding points outside the sample point. Meanwhile, spatial interpolation shows the process of value estimation in the surrounding areas outside the sample point to determine the distribution of values in the area being mapped.

Inverse Weighted Distance (IDW) is used to interpolate prediction results. In IDW, the estimated value of $a$ at location of $x$ shows the average of the weighted closest observations.

$$\hat{a}(x) = \frac{\sum_{i}^{n} h_i j_i}{\sum_{i}^{n} h_i} \tag{9}$$

where $h_i = |x - x_i|^{-\beta}$, $\beta \geq 0$, $|\ldots|$ corresponds to the Euclidean distance and $\beta$ determines the extent to which a point is preferred over other points. If the point $x$ coincides with the observation location or sample point ($x = x_i$), the value of the sample point $x$ is returned to avoid infinity weight.

7. Testing the accuracy of prediction results.

To test the accuracy of prediction results, the MSE, RMSE, and MAE are used. The analysis of the level of vulnerability will be continued when the result of the prediction testing is accurate. On the contrary, if the result of the analysis is not accurate, the analysis of the level of vulnerability will not be continued. It will be analyzed again using machine learning. The calculations of MSE, RMSE, and MAE, which is close to zero, provide good accuracies.

Therefore, the proposed algorithm for detecting tsunami risk areas can be described in Algorithm 1.

Algorithm 1 is the algorithm to detect and determine area affected by tsunami, based on the vegetation indices, using $k$-NN algorithm. Each vegetation index produces one tsunami risk local map, so that Algorithm 1 generates five tsunami risk local maps, *i.e.*, NDVI map, NDWI map, MNDWI map, SAVI map, and NDBI map. Therefore, in this paper, an algorithm to obtain a model that combines all vegetation indices is also proposed. The suitable vegetation indices are selected based on the local map results from Algorithm 1. This model calculates the tsunami risk value for each region and their level

---

**Algorithm 1  Algorithm for detecting tsunami risk areas**

Input: Landsat 8 image data ($X$) and the number of nearest neighbors ($k$).

Step 1: Extract the data into 11 bands.

Step 2: Atmospheric correction, using DOS1 (Eq. 1).

Step 3: Pre-processing computation of vegetation indices, using NDVI (Eq. 2), NDWI (Eq. 3), MNDWI (Eq. 4), SAVI (Eq. 5), and NDBI (Eq. 6).

Step 4: Compute the correlation test, using Pearson correlation (Eq. 7).

Step 5: Forecast and classify the data, using $k$-NN algorithm.

Step 7: Spatial interpolation, using IDW (Eq. 9).

Step 8: Map the tsunami risk areas.

Output: Tsunami risk local map.

---

**Algorithm 2  Algorithm for determining tsunami risk values**

Input: independent variable, $y$, (*i.e.*, level risk value) and dependent variable, $x$, (*i.e.*, vegetation indices and other parameters)

Step 1: Select the dependent variable of vegetation indices based on the results of Algorithm 1.

Step 2: Apply a multiple linear regression, $y = a_0 + a_1x_1 + a_2x_2 + \ldots + a_nx_n$ for the training data.

Step 3: Calculate the risk value for the testing data.

Step 4: Categorize the risk value into level risk.

Output: risk value and level risk

---

risks. Multiple linear regression is used to obtain the model. The algorithm for determining the tsunami risk value and level risk is described in Algorithm 2.

## RESULTS AND DISCUSSIONS

### Data generation

The data used were Landsat 8 OLI image data, which contained 11 bands as listed in Table 1. All 11 bands were processed using QGIS according to the required indices; NDVI, NDBI, MNDWI, NDWI and SAVI. Each index was calculated using Eqs. (2)–(6) to see monthly index results. The data taken for processing were the average result of each index. The data used for this study were yearly data for two different months, August and November, from 47 villages in Kulon Progo Regency. For processing, the data for August and November from 2014 to 2020 were combined. The data were arranged sequentially every year starting from August to November, followed by the index values and classification of each index. In the binary field, each index was adjusted to the potential risk posed and divided into two groups, TRUE (affected by tsunami) and FALSE (unaffected by tsunami). The total binary result of each index was calculated as a sum that indicates TRUE. Since there were five vegetation indices, then each region had a minimum of zero TRUE values and a maximum of five TRUE values. A region was classified into three risk levels, *i.e.*, low risk, medium risk, and high risk. Thus, in this research, it is determined

| Code | Village | Subdistrict | Month | Year | MNDWI | Class_MNDWI | Biner_NDWI | NDBI | Class_NDBI | Biner_NDBI | NDVI | Class_NDVI | Biner_NDVI | SAVI | Class_SAVI | Biner_SAVI |
|------|---------|-------------|-------|------|-------|-------------|------------|------|------------|------------|------|------------|------------|------|------------|------------|
| 1 | Tirtorahayu | Galur | Ags | 14 | -0.113683636 | Negative | FALSE | -0.267873886 | Low Building | FALSE | 0.423695406 | High Vegetation | FALSE | 0.250334571 | Water Body | TRUE |
| 2 | Pandowan | Galur | Ags | 14 | -0.133624775 | Negative | FALSE | -0.232603365 | Low Building | FALSE | 0.39977869 | High Vegetation | FALSE | 0.235702273 | Water Body | TRUE |
| 3 | Brosot | Galur | Ags | 14 | -0.11739655 | Negative | FALSE | -0.234611121 | Low Building | FALSE | 0.394055183 | High Vegetation | FALSE | 0.235803603 | Water Body | TRUE |
| 4 | Karangsewu | Galur | Ags | 14 | -0.180442941 | Negative | FALSE | -0.189978264 | Low Building | FALSE | 0.403557498 | High Vegetation | FALSE | 0.242450261 | Water Body | TRUE |
| 5 | Nomporejo | Galur | Ags | 14 | -0.189308851 | Negative | FALSE | -0.225053675 | Low Building | FALSE | 0.445765956 | High Vegetation | FALSE | 0.270885935 | Water Body | TRUE |
| 6 | Kranggan | Galur | Ags | 14 | -0.127726195 | Negative | FALSE | -0.238212577 | Low Building | FALSE | 0.397963424 | High Vegetation | FALSE | 0.239913419 | Water Body | TRUE |
| 7 | Banaran | Galur | Ags | 14 | -0.158888931 | Negative | FALSE | -0.173422056 | Low Building | FALSE | 0.355732602 | High Vegetation | FALSE | 0.22040389 | Water Body | TRUE |
| 8 | Ngentakrejo | Lendah | Ags | 14 | -0.228313264 | Negative | FALSE | -0.104133792 | Low Building | FALSE | 0.352866227 | High Vegetation | FALSE | 0.221476572 | Water Body | TRUE |
| 9 | Bumirejo | Lendah | Ags | 14 | -0.181545655 | Negative | FALSE | -0.245630879 | Low Building | FALSE | 0.477728224 | High Vegetation | FALSE | 0.292518802 | Water Body | TRUE |
| 10 | Gulurejo | Lendah | Ags | 14 | -0.22599853 | Negative | FALSE | -0.175048159 | Low Building | FALSE | 0.436135425 | High Vegetation | FALSE | 0.278032241 | Water Body | TRUE |
| 11 | Sidorejo | Lendah | Ags | 14 | -0.210477713 | Negative | FALSE | -0.251611881 | Low Building | FALSE | 0.512092307 | High Vegetation | FALSE | 0.324956368 | Water Body | TRUE |
| 12 | Jatirejo | Lendah | Ags | 14 | -0.177740322 | Negative | FALSE | -0.242566069 | Low Building | FALSE | 0.46990353 | High Vegetation | FALSE | 0.285023188 | Water Body | TRUE |
| 13 | Wahyuharjo | Lendah | Ags | 14 | -0.128595963 | Negative | FALSE | -0.236725 | Low Building | FALSE | 0.405894962 | High Vegetation | FALSE | 0.237501492 | Water Body | TRUE |
| 14 | Gulurejo | Lendah | Ags | 14 | -0.210490726 | Negative | FALSE | -0.23578095 | Low Building | FALSE | 0.490729429 | High Vegetation | FALSE | 0.31586667 | Water Body | TRUE |
| 15 | Sidorejo | Lendah | Ags | 14 | -0.182961158 | Negative | FALSE | -0.257163309 | Low Building | FALSE | 0.492976367 | High Vegetation | FALSE | 0.313620342 | Water Body | TRUE |
| 16 | Sendadangsari | Lendah | Ags | 14 | -0.261867434 | Negative | FALSE | -0.177387938 | Low Building | FALSE | 0.456650198 | High Vegetation | FALSE | 0.301415563 | Water Body | TRUE |
| 17 | Ngentakrejo | Lendah | Ags | 14 | -0.222437853 | Negative | FALSE | -0.230771433 | Low Building | FALSE | 0.498312137 | High Vegetation | FALSE | 0.317370885 | Water Body | TRUE |
| 18 | Krembangan | Panjatan | Ags | 14 | -0.215802013 | Negative | FALSE | -0.252987413 | Low Building | FALSE | 0.519000752 | High Vegetation | FALSE | 0.325243217 | Water Body | TRUE |
| 19 | Cerme | Panjatan | Ags | 14 | -0.224602919 | Negative | FALSE | -0.272899267 | Low Building | FALSE | 0.549492544 | High Vegetation | FALSE | 0.341841331 | Water Body | TRUE |
| 20 | Gotakan | Panjatan | Ags | 14 | -0.210954366 | Negative | FALSE | -0.25279081 | Low Building | FALSE | 0.518485258 | High Vegetation | FALSE | 0.318002723 | Water Body | TRUE |

**Figure 3  Calculation results of each index.**                                 

and simulated that if the number of TRUE ≤ 1, a region has a low potential/impact (low risk), if the number of TRUE = 2, a region has a moderate potential/impact (medium risk), and if the number of TRUE ≥ 3, a region has a high potential/impact of tsunami (high risk). Figure 3 shows the calculation results of each index and their resulting potentials.

## Correlation between variables
Based on the results of the Pearson correlation analysis, it showed that the correlation between the vegetation indices of the three tsunami impact risk classes had a positive or negative correlation value. As presented in Fig. 4, it can be seen that the lowest correlation with a negative value was the correlation between the MNDWI and NDVI indices, with a correlation value of −0.761. The scatter diagram showed the increase in the MNDWI index value which was not in line with the increase in the NDVI value for the potential tsunami risk. The highest correlation that had a positive value was the correlation between the SAVI and NDVI indices of 0.767. The distribution pattern of the data contained in the scatter diagram was closer to a straight line and the risk of tsunami impact increased in line with the increase in the value of the vegetation indices. The positive correlation between these two indices could be seen as the strongest relationship in the medium risk of tsunami impact. A positive correlation was indicated by the distribution pattern of the data pair points that moved closer to a straight line, which showed a close relationship between the risk of the tsunami impact and the index. This relationship might also be referred to as a unidirectional relationship. The visualization of the data displayed was not only in the form of closeness value between variables but also in the form of a bar plot presenting the distribution of data from 2014 to 2020. Data illustrated that moderate tsunami risks are more dominant than low tsunami risks.

## Prediction and classification (k-NN)
Prediction and classification were done using the $k$-NN algorithm, where the number of $k$ was determined by classification using $k$-NN with training data 70% and testing data 30%
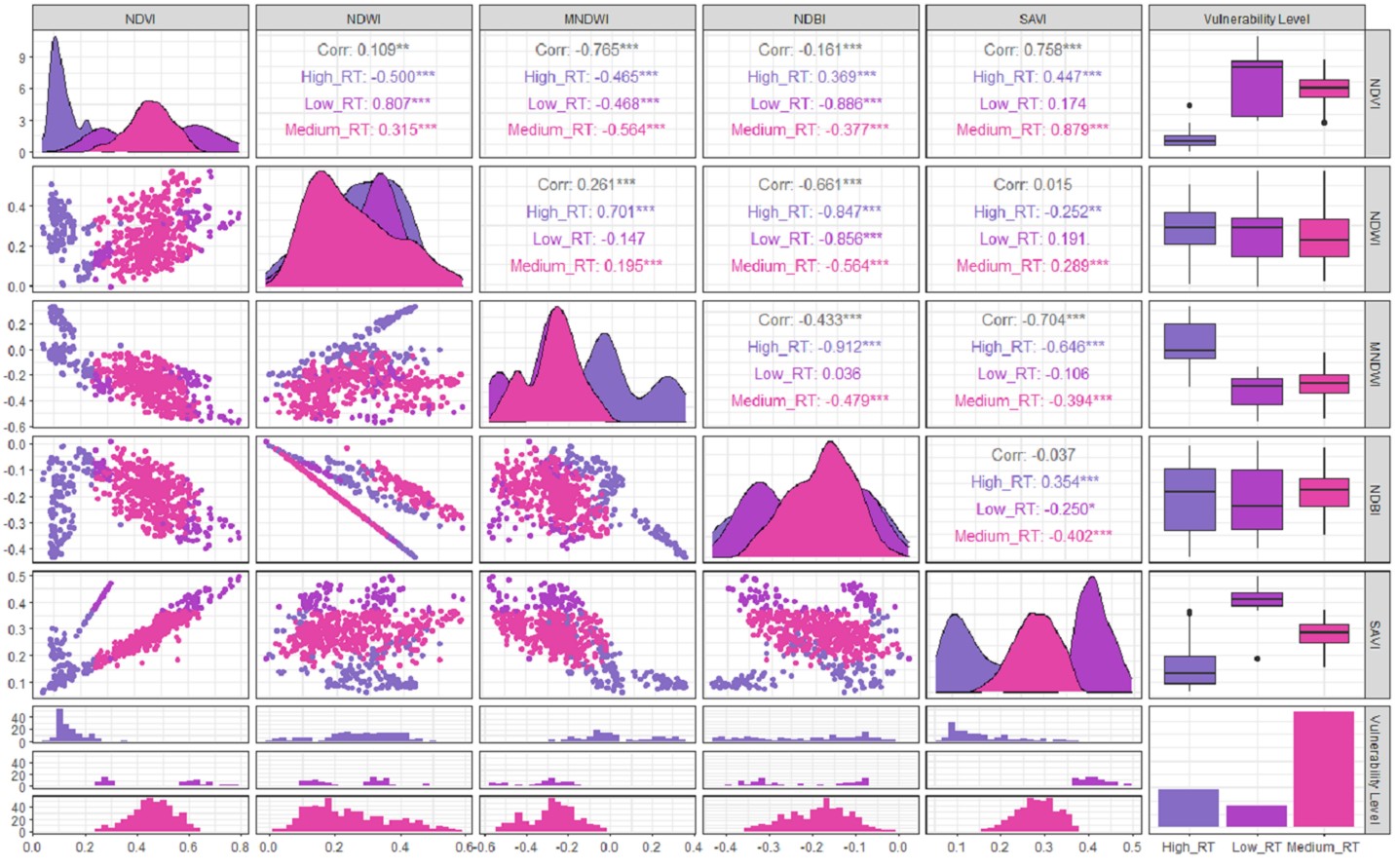

**Figure 4 Correlation between variables.**

of the total data. To compare the performance of $k$, the calculation of the accuracy and value of the Kappa coefficient was carried out starting from $k = 1$ up to $k = 15$. Based on the obtained results in Fig. 5, the highest accuracy and value of Kappa was in $k$-NN with $k = 1$ and 2. Therefore, in this study, predictions were made using $k = 1$.

The testing of performance modeling is carried out by calculating Mean Squared Error (MSE), Mean Absolute Error (MAE), Root Mean Squared Error (RMSE), and Cohen's Kappa. MSE is the sum of the squared error or the difference between the actual value and the predicted value, where $MSE = \sum \frac{(Y' - Y)^2}{n}$, with $Y'$ is the predicted value, $Y$ is the actual value, and $n$ is the total data. MAE shows the average error value from the actual value compared to the predicted value, where $MAE = \sum \frac{|Y' - Y|}{n}$. Usually, it is used for measuring error prediction in time series analysis. RMSE is the square root of MSE, where $RMSE = \sqrt{\sum \frac{(Y' - Y)^2}{n}}$. Cohen's Kappa is used to measure the degree of agreement of

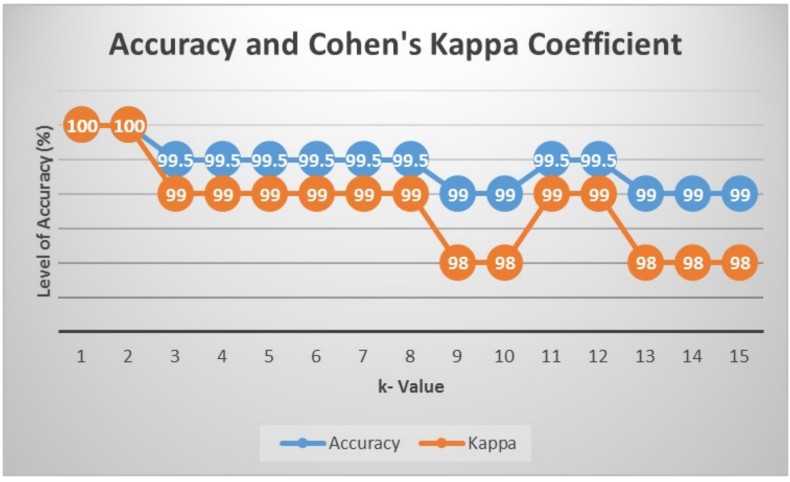

**Figure 5 Accuracy and value of Cohen's kappa.**

**Table 2 MSE, RMSE, and MAE values of each index with prediction results using *k*-NN, ANN, SVM, and cart algorithms.**

| | MSE | | | | RMSE | | | | MAE | | | |
|---|---|---|---|---|---|---|---|---|---|---|---|---|
| Algorithm | *k*-NN | ANN | SVM | Cart | *k*-NN | ANN | SVM | Cart | *k*-NN | ANN | SVM | Cart |
| **Index** | | | | | | | | | | | | |
| MNDWI | 0.0015 | 0.0015 | 0.0013 | 0.0015 | 0.0147 | 0.0387 | 0.0363 | 0.0388 | 0.0070 | 0.0308 | 0.0256 | 0.0309 |
| SAVI | 0.0002 | 0.0006 | 0.0007 | 0.0007 | 0.0130 | 0.0244 | 0.0266 | 0.0266 | 0.0063 | 0.0199 | 0.0191 | 0.0212 |
| NDVI | 0.0006 | 0.0015 | 0.0015 | 0.0018 | 0.0244 | 0.0385 | 0.0384 | 0.0423 | 0.0106 | 0.0302 | 0.0260 | 0.0335 |
| NDWI | 0.0003 | 0.0007 | 0.0006 | 0.0008 | 0.0180 | 0.0262 | 0.0253 | 0.0286 | 0.0086 | 0.0191 | 0.0166 | 0.0218 |
| NDBI | 0.0003 | 0.0008 | 0.0007 | 0.0008 | 0.0184 | 0.0281 | 0.0258 | 0.0286 | 0.0086 | 0.0214 | 0.0174 | 0.0218 |

the two raters in classifying objects into groups and in measuring the agreement between the new methods and the existing methods, where *Cohen's Kappa* $= \dfrac{\Pr(a) - \Pr(e)}{1 - \Pr(e)}$, with $\Pr(a)$ is the percentage of consistent number of measurements between raters and $\Pr(e)$ is the percentage of change in measurement between raters.

After the predictions are done, the MSE, RMSE, and MAE are calculated to see the prediction performance. Table 2 shows the test of error and the resulting average from data using the prediction results of each index with *k*-NN, ANN, SVM, and CART algorithms.

The values of MSE, RMSE, and MAE for *k*-NN, ANN, SVM, and Cart of each index can be shown in Figs. 6–8, respectively. As can be seen from these figures, the *k*-NN algorithm obtains the best performance for all indices, compared with ANN, SVM, and Cart algorithms.

Furthermore, the averages of each index using all algorithms are compared to the actual data. The results can be seen in Table 3. From this table, k-NN obtains the closest average value for each index.
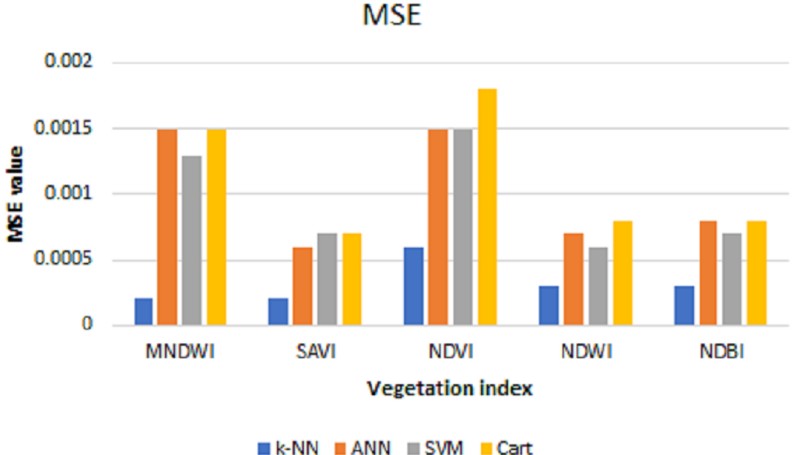

**Figure 6**  MSE values for *k*-NN, ANN, SVM, and cart.

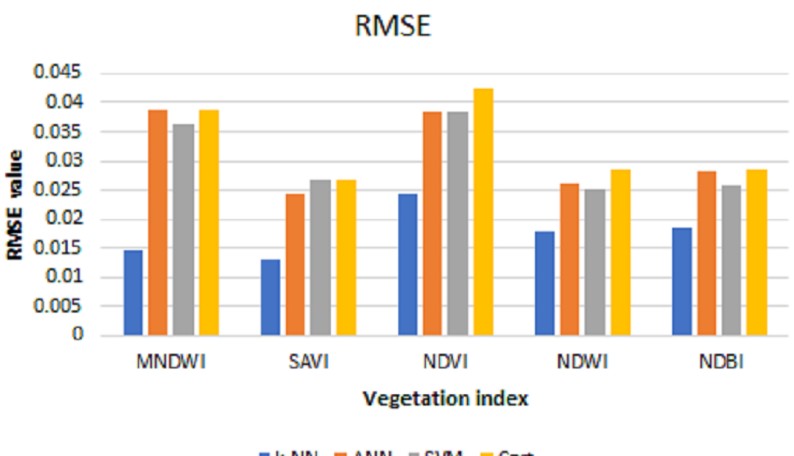

**Figure 7**  RMSE values for *k*-NN, ANN, SVM, and cart.

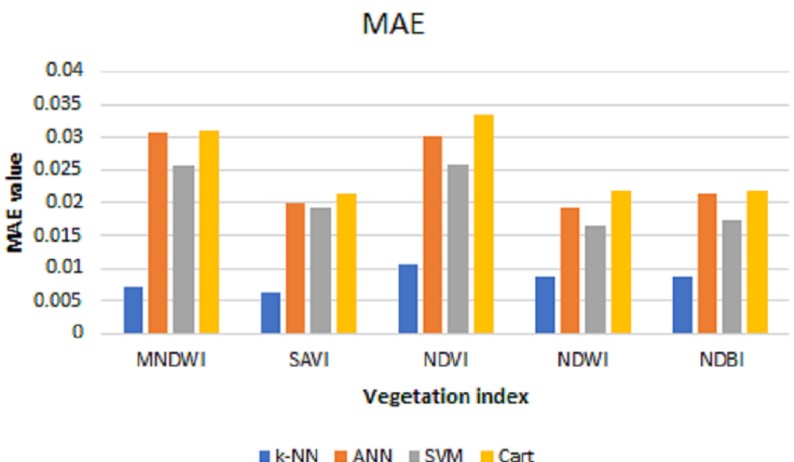

**Figure 8**  MAE values for *k*-NN, ANN, SVM, and cart.

**Table 3 The average of each index using k-NN, ANN, SVM, and cart.**

|  | MNDWI | SAVI | NDVI | NDWI | NDBI |
|---|---|---|---|---|---|
| k-NN | −0.3411 | 0.2928 | 0.4873 | 0.2058 | −0.2055 |
| ANN | −0.3308 | 0.2909 | 0.4841 | 0.2024 | −0.2018 |
| SVM | −0.3305 | 0.2867 | 0.4809 | 0.2000 | −0.1991 |
| Cart | −0.3305 | 0.2907 | 0.4838 | 0.2056 | −0.2056 |
| Actual | −0.3406 | 0.2962 | 0.4941 | 0.2055 | −0.2055 |

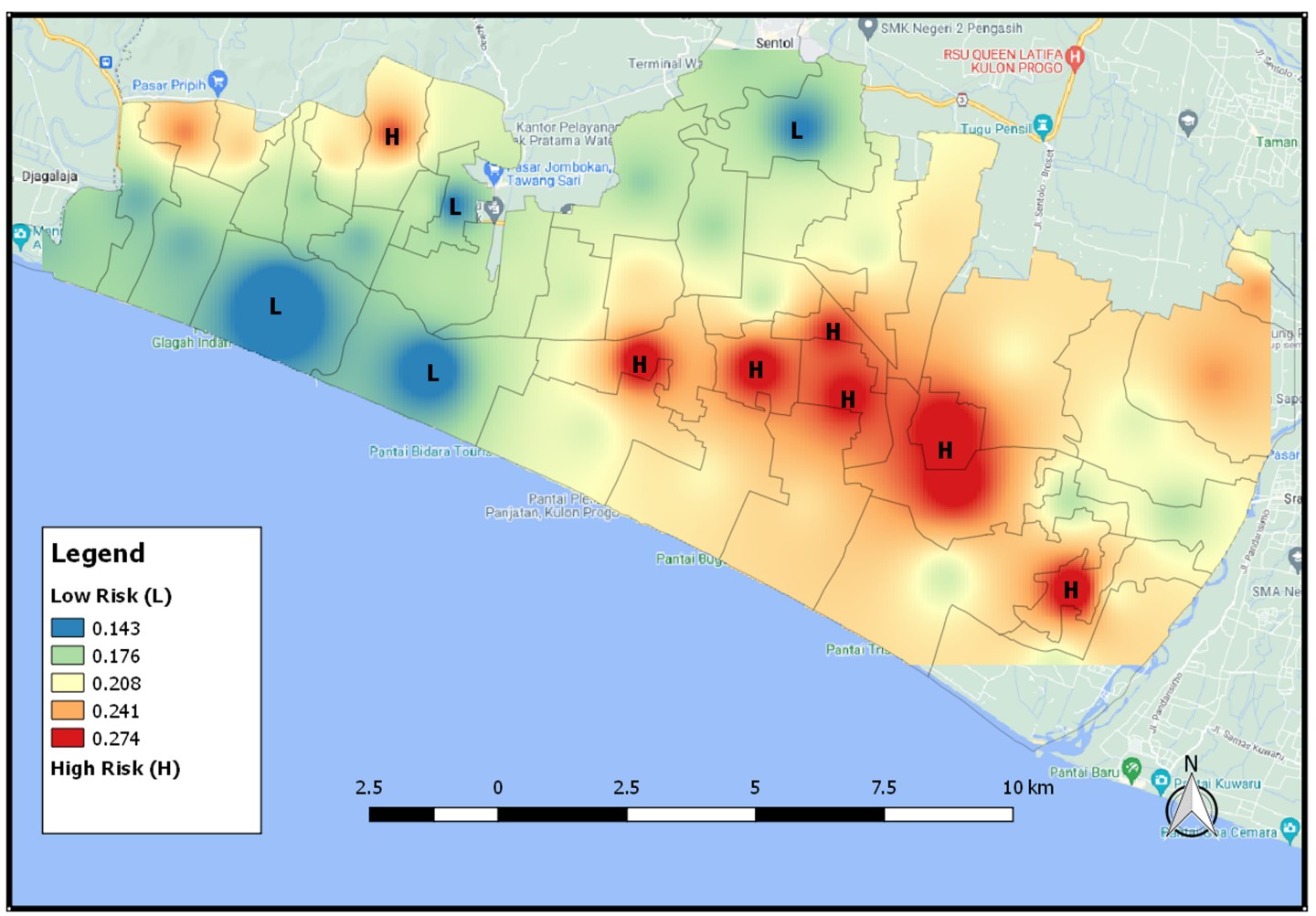

**Figure 9 The distribution map of tsunami risk areas based for NDWI index 2021.**

## The map of tsunami risk areas

Figure 9 is the map showing the distribution of areas that have the potential for a tsunami using the IDW interpolation results. The higher the level of vulnerability of the area affected by the tsunami, the redder the area will be and the lower the level of vulnerability, the bluer the color will be. An area will be classified as a high vulnerability area when it has

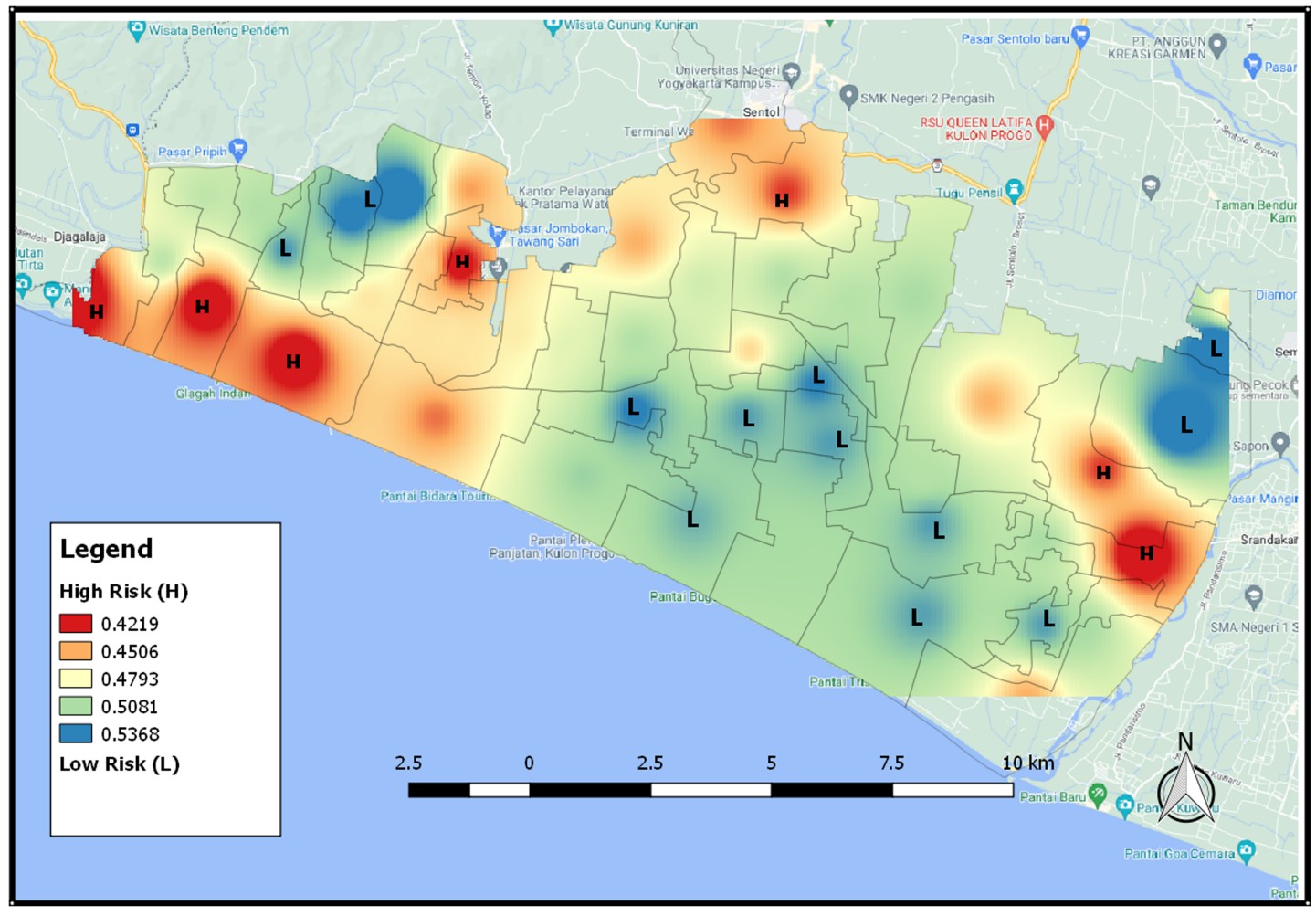

**Figure 10 The distribution map of tsunami risk areas based for NDVI 2021 index.**

a high wettability value and a few vegetation. Figure 9 shows the distribution map of tsunami risk areas based on NDWI index 2021. When an area is dominated by water or has a high level of wetness, it will have a high level of tsunami risk so that the area has a reddish color. On the contrary, an area with a low level of tsunami risk has a bluish color.

The map of the distribution of potential areas in Fig. 10 provides tsunami risk areas based on the NDVI index. The higher the NDVI value, the higher possibility of an area having high vegetation. An area that has a low risk of tsunami, its color will be bluish because it has a lot of vegetation. For an area with a high risk of tsunami, its color will be close to red because the area has a low NDVI value and thus it shows that the area has only a few vegetation.

The SAVI index presents areas that are adjusted according to their type of soil or the shape of the areas. Higher SAVI value indicates that the area is in the form of forested vegetation, while lower value means the area has many bodies of water such as rivers. Figure 11 shows that the area that is colored redder, which means that the area has a low

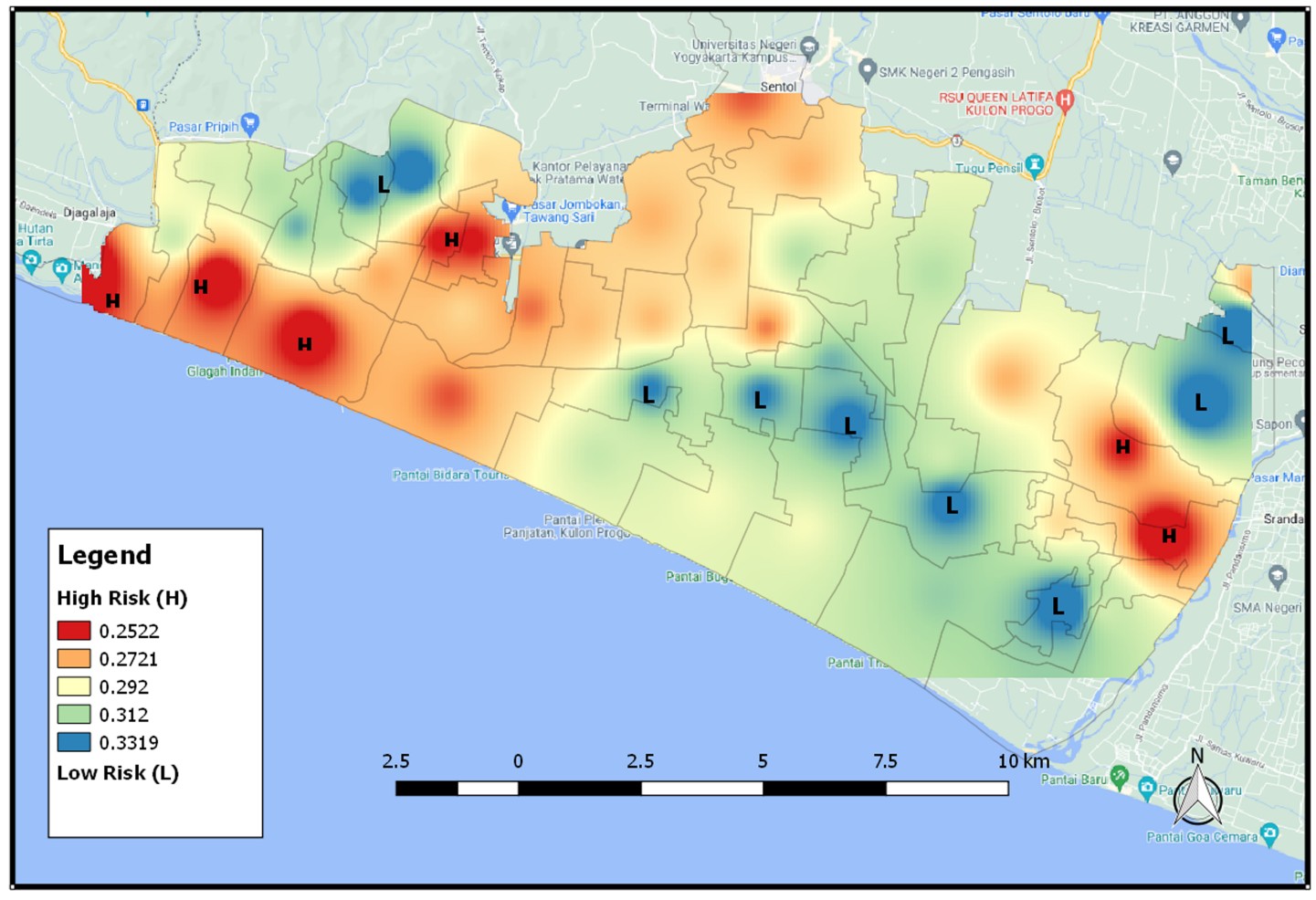

**Figure 11 The distribution map of tsunami risk areas for SAVI index 2021.**

SAVI value and has a high level of tsunami vulnerability because it has many areas of water, asphalt, paving, etc., meanwhile the blue area in the figure is an area that has a low level of tsunami vulnerability. Figure 11 also shows the distribution map of tsunami risk areas based on SAVI index 2021.

Figure 12 is a slope map that describes the slope angle in the study area of Kulon Progo Regency. The slope map is represented by the percentage of the slope angle, *i.e.*, 0–8%, 8–15%, 15–25%, 25–40%, and >40%. This research analyzes the slope as one of the reference data for making maps of tsunami-prone areas. The results of the spatial analysis suggest that most of the study areas have undulating slopes, some are moderately sloping and some are hilly. The quantitative value of the slope map is used as an indicator to determine areas having a high risk of being prone to tsunami.

Furthermore, since there are only three categories of risk level, they are high, medium, and low risk and no value that indicates the level of tsunami risk, then a new formula is also proposed and discussed to determine the tsunami risk value. In this formula, vegetation indices, slope, and distance are used to obtain the risk value, where distance is

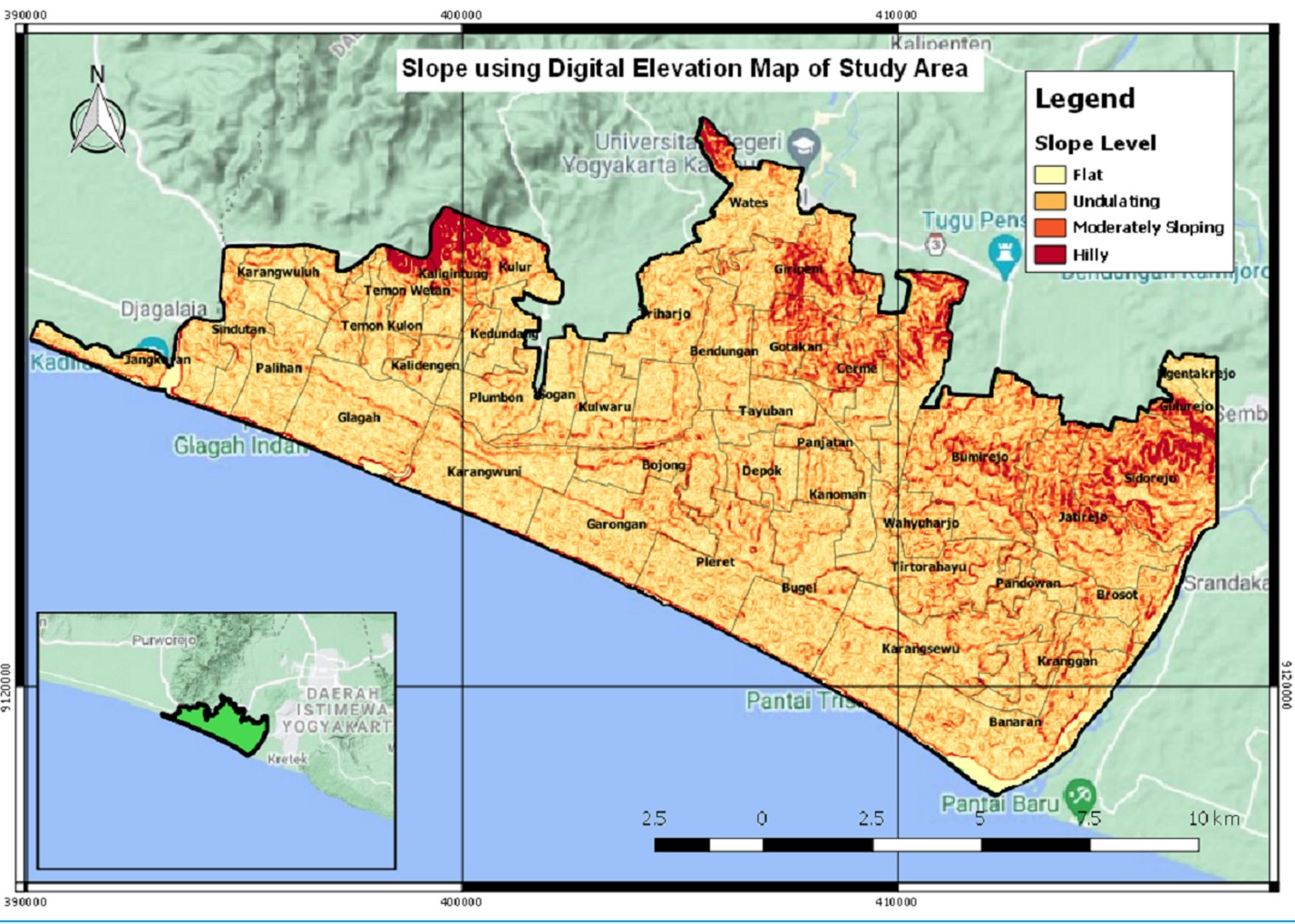

**Figure 12 The slope map.**

the shortest distance of a region with the shoreline. The value is on interval [0,1]. The higher the risk value, *RV*, the higher the area is at risk of a tsunami.

In this experiment, the data of vegetation indices, slope, and distance from 47 villages of 2014 until 2019 were used as the training data. NDBI and MNDWI were not used in the formula since it had no effect in determining the risk area, based on the experiment using *k*-NN. Since a region with low slope value, low NDVI value, or low distance had a high tsunami risk level, then they were applied as denominators in the model. Therefore, according to slope, NDWI, NDVI, and SAVI values of the training data as the independent variables and level risk values of the training data as the dependent variables, Algorithm 2 can be used to obtain the risk values of the testing data. With multiple linear regression model using Excel, the tsunami risk level, *RV*, was modeled as in Eq. (10) as below,

$$RV = \frac{0.95}{slope} + 0.044 * NDWI - \frac{0.007}{NDVI} + 0.3 * SAVI + \frac{0.59}{distance} \tag{10}$$

**Table 4 Prediction of 2021 tsunami risk values.**

| Village | Risk level | Category |
|---|---|---|
| Banaran | 0.943 | High risk |
| Bendungan | 0.486 | Medium risk |
| Bojong | 0.626 | Medium risk |
| Brosot | 0.401 | Medium risk |
| Bugel | 0.959 | High risk |
| Bumirejo | 0.388 | Low risk |
| Cerme | 0.382 | Low risk |
| Demen | 0.541 | Medium risk |
| Depok | 0.576 | Medium risk |
| Garongan | 0.953 | High risk |
| Giripeni | 0.297 | Low risk |
| Glagah | 0.965 | High risk |
| Gotakan | 0.350 | Low risk |
| Gulurejo | 0.282 | Low risk |
| Jangkaran | 0.897 | High risk |
| Janten | 0.546 | Medium risk |
| Jatirejo | 0.341 | Low risk |
| Kalidengen | 0.671 | Medium risk |
| Kaligintung | 0.353 | Low risk |
| Kanoman | 0.583 | Medium risk |
| Karangsewu | 0.937 | High risk |
| Karangwuluh | 0.625 | Medium risk |
| Karangwuni | 0.967 | High risk |
| Kebonrejo | 0.675 | Medium risk |
| Kedundang | 0.590 | Medium risk |
| Kranggan | 0.784 | Medium risk |
| Krembangan | 0.367 | Low risk |
| Kulur | 0.354 | Low risk |
| Kulwaru | 0.640 | Medium risk |
| Ngentakrejo | 0.555 | Medium risk |
| Ngestiharjo | 0.558 | Medium risk |
| Nomporejo | 0.663 | Medium risk |
| Palihan | 0.965 | High risk |
| Pandowan | 0.460 | Medium risk |
| Panjatan | 0.927 | High risk |
| Pleret | 0.918 | High risk |
| Plumbon | 0.768 | Medium risk |
| Sidorejo | 0.309 | Low risk |
| Sindutan | 0.955 | High risk |
| Sogan | 0.752 | Medium risk |
| Tayuban | 0.531 | Medium risk |

| Village | Risk level | Category |
|---|---|---|
| Temon Kulon | 0.664 | Medium risk |
| Temon Wetan | 0.410 | Medium risk |
| Tirtorahayu | 0.607 | Medium risk |
| Triharjo | 0.495 | Medium risk |
| Wahyuharjo | 0.484 | Medium risk |
| Wates | 0.860 | High risk |

with $R^2$ of 0.921, meaning 92.1% of the variation in the dependent variable (*i.e.*, risk value) could be explained by the independent variables (*i.e.*, slope, NDWI, NDVI, SAVI, and distance).

To test the model, data from 2020 were used as the testing data. The MSE of the testing data was 0.047. Besides that, the risk level was categorized into low, medium, and high risk. Low risk was in interval [0, 0.4) of risk value, medium risk was in interval [0.4, 0.8) of risk value, and high risk was in interval [0.8, 1] of risk value. The model obtained an accuracy rate of 93.62%. Using the model from Eq. (10), the prediction of 2021 tsunami risk values for 47 villages is listed in Table 4.

## CONCLUSIONS

To sum up, the paper proposed a new algorithm to predict tsunami risk areas based on spatial prediction of vegetation indices. Atmospheric correction using the DOS1 algorithm is used for image correction since the field data parameters are not known. $k$-NN is used to classify and predict tsunami-affected areas from vegetation indices data that has spatial and temporal resolutions. Experimental results and comparisons demonstrate the effectiveness of the proposed algorithm to detect the tsunami risk area. The result with $k$-NN algorithm gives MSEs of 0.0002 for MNDWI, 0.0002 for SAVI, 0.0006 for NDVI, 0.0003 for NDWI, and 0.0003 for NDBI. Moreover, based on the tsunami risk area of each vegetation index, a prediction model also has been proposed using NDVI, NDWI, SAVI, slope, and distance to obtain the tsunami risk value of an area, by using multiple linear regression with $R^2$ of 0.921. The accuracy rate from categorizing the risk value into level risk is about 93.62%. For future work, besides vegetation indices, slope, and distance, other parameters can also be used to find the tsunami risk area with a more accurate risk value, such as elevation, sea level, etc. Parameters that are not related to the determination of a potentially tsunami-prone area may also be found by certain methods. Another future work is to compare the Euclidean distance used in $k$-NN algorithm with other numerical similarity metric, such as cosine distance, Jaccard distance, etc.

### Funding

This work was supported by the Education and Culture Ministry Republic Indonesia for Grant Research World Class Professor at 2021. The funders had no role in

study design, data collection and analysis, decision to publish, or preparation of the manuscript.

## Grant Disclosures

The following grant information was disclosed by the authors:
Education and Culture Ministry Republic Indonesia.

## Competing Interests

The authors declare that they have no competing interests.

## Author Contributions

- Kristoko Dwi Hartomo conceived and designed the experiments, performed the experiments, analyzed the data, performed the computation work, prepared figures and/or tables, authored or reviewed drafts of the paper, and approved the final draft.
- Yessica Nataliani conceived and designed the experiments, performed the experiments, analyzed the data, performed the computation work, prepared figures and/or tables, and approved the final draft.
- Zainal Arifin Hasibuan analyzed the data, authored or reviewed drafts of the paper, and approved the final draft.

## Data Availability

The data is available at GitHub: https://github.com/yessno24/tsunami.

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
