# Peer review of "Vegetation indices’ spatial prediction based novel algorithm for determining tsunami risk areas and risk values"

_PeerJ Computer Science, doi:10.7717/peerj-cs.935_

## Round 0.1 · original submission · Minor Revisions

The paper is interesting and addresses a critical problem. It is well written and describes most of the aspects. However, considering the reviewers' comments, I recommend a minor review of the paper and invite you to submit a revised manuscript after addressing all reviewers' comments. Also, thoroughly revise the manuscript to correct all spelling/grammatical/typo/style errors. Furthermore, consider revising the title to "Vegetation indeces' spatial prediction based novel algorithm for determining tsunami risk areas and risk values".

·

Basic reporting

1. There is a need to check the grammar, spelling, sentence structures, etc. of the paper.
In the title “A New Algorithm for …….on Spatial Predicition of 3 Vegetation Index”. The word “Index” should be in plural form since more than one index is used in the paper.
In other parts of the paper, I found several errors on the spelling of words, verb agreement, correct usage of articles, etc. There is a need to go over the entire paper to check on these errors and reconstruct or rephrase some sentences which are redundant (e.g.Lines 239-242 needs to be reconstructed to avoid redundancy). I would suggest that the authors use an APP to correct these errors.
2. A reorganization of the Abstract is recommended. The results of the algorithm should be written in the last lines and the presentation of the algorithm and the indexes used should be more profound to emphasize the objective of the paper.
3. The tsunami risk levels (Levels 1, 2, and 3) are based on indicators resulting from the algorithm (Lines 358-360). First, the basis of these indicators was not explicitly explained. Second, how would you classify the risk if TRUE is within the range (1,2) or (2,3)? Is it always the case that the algorithm will result to TRUE=2 in your simulations?
4. In your regression model (Equation 10), the authors need to explain how they obtained the constant coefficients (e.g. 0.95, 0.007, etc.)
5. Comments on Figures
Are the location maps taken by the authors themselves? If not, then there is a need to indicate the source below the figure. Figure 3, although it’s a captured image, its’ source should be written below. Also, the figure is more likely to be identified as a table rather than a figure. The x-axis and y-axis must be given labels on Figures 5,6,7,8 and the titles whould be more clear and concise. In Figures 9,10,11, the legend uses a color coding to show the level of risk is not properly defined. Since you are identifying the risk levels with color codes, some colors are not properly identified. Also, it would be appropriate to give the range of values for the risk levels so that readers of the paper can properly distinguish the different risk levels in the graph.
6. Comments on Tables
Since the purpose of the authors is to make a comparative analysis on the amount of error generated using different algorithms and that of their proposed algorithm, I would suggest either of the following improvements:
a. Tables 2,3,4,5 can be reconstructed as one table showing the amount of error (MSE. RMSE, MAE) of each algorithm based on each index, or
b. Three separate tables for each classified error (i.e. MSE. RMSE, MAE) comparing the algorithms and the amount of error based on the 3 indexes.
Since each table enumerates the amount of error of each index with prediction results using the varied machine learning algorithms in your study, it would be better to fuse all 4 tables as one where all the algorithms are listed together with the indexes and the classification of errors. In this way, you can make a clear comparison of the amount of error. Alternatively, you may create three tables, each for the identified error MSE, RMSE, MAE separately, and compare the results of the algorithms.

Experimental design

the experimental design is well explained and detailed

Validity of the findings

The findings are valid based on the provided data. The objectives of the study were achieved.

Reviewer 2 ·

Basic reporting

This paper studies a new algorithm to detect tsunami risk areas based on spatial modelling of vegetation index. I think three questions must be answered in this paper.

1. I suggest to add some more explanation for the flowchart/algorithm to make it more clear and understandable.
2. In Eq. (8), the closest point search technique is performed by using the Euclidean distance formula. Why did you prefer Euclidean distance instead of Hamming distance or Hausdorff distance?
3. Is it possible that the closest point search technique is performed by using the similarity between data points instead of Euclidean distance?

Experimental design

No comments

Validity of the findings

No comments

Annotated reviews are not available for download in order to protect the identity of reviewers who chose to remain anonymous.

---

## Round 0.2 · accepted · Accept

I congratulate the authors for addressing the reviewers' comments well . The paper is in excellent shape now. I recommend authors thoroughly review the manuscript to correct all typo/spelling/grammatical/style errors.

·

Basic reporting

The English structure of the revised paper shows clarity and a more improved version than the initial paper submitted. Figures and Tables are given the appropriate titles and the data of their results and the raw data were shared. All terms and Acronyms used in this paper are defined according to the context of their paper

Experimental design

no comment
Everything is accounted for

Validity of the findings

Findings are valid and conclusions are well stated with future recommendations

Additional comments

Dear Peer J Team,
By comparing my suggested comments and revisions of my previous review of this paper and the newly submitted revised paper, all my comments and suggestions were addressed which improved a lot the quality and consistency of this paper.

Reviewer 2 ·

Basic reporting

The authors have addressed my concerns

Experimental design

ok

Validity of the findings

ok

Additional comments

n/a